# Mass Spectrometric Profiling of Extraocular Muscle and Proteomic Adaptations in the *mdx-4cv* Model of Duchenne Muscular Dystrophy

**DOI:** 10.3390/life11070595

**Published:** 2021-06-22

**Authors:** Stephen Gargan, Paul Dowling, Margit Zweyer, Jens Reimann, Michael Henry, Paula Meleady, Dieter Swandulla, Kay Ohlendieck

**Affiliations:** 1Department of Biology, National University of Ireland, Co. Kildare, W23F2H6 Maynooth, Ireland; stephen.gargan@mu.ie (S.G.); paul.dowling@mu.ie (P.D.); 2Kathleen Lonsdale Institute for Human Health Research, Maynooth University, Co. Kildare, W23F2H6 Maynooth, Ireland; 3Department of Neonatology and Paediatric Intensive Care, Children’s Hospital, University of Bonn, D53113 Bonn, Germany; margit.zweyer@dzne.de; 4Department of Neurology, University Hospital, University of Bonn, D53113 Bonn, Germany; jens.reimann@ukbonn.de; 5National Institute for Cellular Biotechnology, Dublin City University, D09V209 Dublin, Ireland; michael.henry@dcu.ie (M.H.); paula.meleady@dcu.ie (P.M.); 6Institute of Physiology II, University of Bonn, D53115 Bonn, Germany; dieter.swandulla@ukbonn.de

**Keywords:** Duchenne muscular dystrophy, dystrophinopathy, extraocular muscle, glyceraldehyde-3-phosphate dehydrogenase, myosin-14, myosin heavy chain

## Abstract

Extraocular muscles (EOMs) represent a specialized type of contractile tissue with unique cellular, physiological, and biochemical properties. In Duchenne muscular dystrophy, EOMs stay functionally unaffected in the course of disease progression. Therefore, it was of interest to determine their proteomic profile in dystrophinopathy. The proteomic survey of wild type mice and the dystrophic *mdx-4cv* model revealed a broad spectrum of sarcomere-associated proteoforms, including components of the thick filament, thin filament, M-band and Z-disk, as well as a variety of muscle-specific markers. Interestingly, the mass spectrometric analysis revealed unusual expression levels of contractile proteins, especially isoforms of myosin heavy chain. As compared to diaphragm muscle, both proteomics and immunoblotting established isoform MyHC14 as a new potential marker in wild type EOMs, in addition to the previously identified isoforms MyHC13 and MyHC15. Comparative proteomics was employed to establish alterations in the protein expression profile between normal EOMs and dystrophin-lacking EOMs. The analysis of *mdx-4cv* EOMs identified elevated levels of glycolytic enzymes and molecular chaperones, as well as decreases in mitochondrial enzymes. These findings suggest a process of adaptation in dystrophin-deficient EOMs via a bioenergetic shift to more glycolytic metabolism, as well as an efficient cellular stress response in EOMs in dystrophinopathy.

## 1. Introduction

As one of the most abundant cellular entities in the human body, the diverse types of skeletal muscle fibre form the contractile units of over 650 individual muscles. Voluntary striated muscles differ considerably in their histological, anatomical, metabolic, biochemical, and physiological specialization between predominantly slow-twitching versus fast-twitching phenotypes [1]. A distinctly different subtype of skeletal muscles, as compared to non-craniofacial muscles, is presented by extraocular muscles (EOMs) [2,3], which control the movements of the eyes [4]. The finely tuned and highly coordinated actions of six EOMs, i.e., *M. obliquus superior, M. obliquus inferior*, *M. rectus medialis, M. rectus lateralis, M. rectus superior*, and *M. rectus inferior* [5], provide the eyeball with a wide range of complex movements about its horizontal and vertical axis. This includes a diverse range of eye movements ranging from gaze-holding to slow vergence, smooth pursuit convergence, optokinetic responses, vestibulo-ocular reflexes, and rapid saccades [6].

Characteristic biological features of EOMs include (i) unique developmental processes with specific upstream activators [7,8], (ii) longitudinal dispersion of multiterminal neuromuscular junctions [9,10,11], (iii) morphologically distinct muscle spindles as compared to conventional somatic spindles [12], (iv) an unusually high capacity for cellular remodelling and fibre regeneration [13,14], (v) an efficient calcium handling and extrusion system [15], (vi) high levels of fatigue resistance even in extremely fast-twitching fibres [16] and (vii) distinguishing combinations of contractile protein isoform expression patterns [17]. Another striking biomedical feature of EOMs is their apparent resistance to degeneration in X-linked muscular dystrophy [18,19,20].

Duchenne muscular dystrophy is an X-chromosomally inherited and severely debilitating disorder of early childhood that primarily affects contractile tissues [21]. Mutations in the *DMD* gene result in the almost complete loss of the full-length isoform of dystrophin, which exhibits high levels of sequence similarity with membrane cytoskeletal proteins, and the concomitant reduction in a variety of sarcolemmal glycoproteins [22]. The extremely large *DMD* gene contains seven different promoters that are involved in the tissue-specific expression of several small dystrophin isoforms, i.e., Dp45 (central nervous system), Dp71-G (ubiquitous), Dp116-S (Schwann cells), Dp140-B/K (brain and kidney) and Dp260-R (retina), as well as the large dystrophins, i.e., Dp427-M (skeletal muscle, heart muscle, smooth muscle), Dp427-B (brain) and Dp427-P (Purkinje cells) [23].

The collapse of the dystrophin-associated glycoprotein complex, which acts as a key signaling hub in normal muscles [23,24], causes weakening of the actin cytoskeleton–sarcolemma–extracellular matrix axis. This results in micro-rupturing events at the level of the sarcolemmal membrane system and triggers abnormal lateral force transmission, impaired cellular signaling and enhanced levels of calcium-induced proteolysis [25,26]. These pathophysiological changes affect the majority of the skeletal musculature, which undergoes cellular degeneration, partial replacement by fatty tissue, severe fibrotic scarring and chronic inflammation [27,28]. In addition, respiratory deficiencies, late-onset cardiomyopathy and scoliosis, as well as neuronal and metabolic complications, are characteristic features of the complex pathology of X-linked muscular dystrophy [26,29,30].

In contrast to other types of skeletal muscle, as previously reviewed in detail [31,32,33], relatively limited numbers of studies on the normal EOM proteome [34,35,36] or proteome-wide changes in dystrophin-deficient EOMs [37,38] have been carried out. We therefore performed a systematic mass spectrometric analysis of EOM preparations and potential adaptations in the dystrophic *mdx-4cv* phenotype, as compared to the recently established proteomic changes in the severely affected *mdx-4cv* diaphragm [39]. Initially, the proteomic profile of wild type EOMs was determined with the help of an Orbitrap Fusion Tribrid mass spectrometer, which identified unusual expression levels of contractile proteins, especially isoforms of myosin heavy chain. Subsequently, comparative proteomics was used to identify changes in the protein expression profile between unaffected EOMs and dystrophin-lacking EOMs. The mass spectrometric characterization of *mdx-4cv* EOMs showed increases in a variety of proteins, including glycolytic enzymes and molecular chaperones.

## 2. Materials and Methods

### 2.1. Materials

For the mass spectrometric analysis of extraocular muscle preparations, analytical grade reagents and general materials were obtained from Sigma Chemical Company (Dorset, UK), GE Healthcare (Little Chalfont, Buckinghamshire, UK) and Bio-Rad Laboratories (Hemel-Hempstead, Hertfordshire, UK). MS-grade trypsin protease was obtained from ThermoFisher Scientific (Dublin, Ireland), as was the Pierce 660 nm Protein Assay Reagent. Spin filters of the type Vivacon 500 (VN0H22; 30,000 MWCO) were purchased from Sartorius (Göttingen, Germany) for carrying out filter-aided sample preparations. Gel electrophoretic separation and immunoblotting was performed with precast Invitrogen Bolt 4–12% Bis-Tris gels and Whatman nitrocellulose transfer membranes from Bio-Science Ltd. (Dun Laoghaire, Ireland), respectively. InstantBlue Coomassie Protein Stain was obtained from Expedeon (Heidelberg, Germany). For immunoblot analysis and immunofluorescence microscopy, primary antibodies were obtained from R&D Systems, Minneapolis, MN, USA (MAB5718 against glyceraldehyde-3-phosphate dehydrogenase), ProteinTech, Rosemont, IL, USA (20716-1-AP against myosin heavy chain 14) and Abcam, Cambridge, UK (ab2413 against fibronectin). Secondary peroxidase-conjugated anti-mouse IgG and anti-rabbit IgG were purchased from Sigma Chemical Company (Dorset, UK) and Cell Signalling Technology (Danvers, MA, USA), respectively.

### 2.2. Extraocular Muscle Specimens

The harvesting of post mortem EOMs and diaphragm muscle specimens from 12-month-old wild type mice and the *mdx-4cv* mouse model of dystrophinopathy, which lacked the dystrophin isoforms Dp140, Dp260 and Dp427 due to a point mutation in exon 53, was carried out according to institutional regulations. Animals were handled in strict adherence to local governmental and institutional animal care regulations and were approved by the Institutional Animal Care and Use Committee (Amt für Umwelt, Verbraucherschutz und Lokale Agenda der Stadt Bonn, North Rhine-Westphalia, Germany). Frozen specimens were transported on dry ice to Maynooth University in accordance with the regulations of the Department of Agriculture (animal by-product register number 2016/16 to the Department of Biology, National University of Ireland, Maynooth). The eyeball and its surrounding tissues were carefully removed from the ocular cavity by bulbar exenteration. The EOM cone excluding the *retractor bulbi* muscle was then dissected out and extracted for the isolation of the combined EOM proteome. The establishment of multi-consensus files was carried out with muscle samples from 6 wild type and 6 dystrophic mice. Comparative proteomics was performed with specimens from 3 wild type versus 3 dystrophic mice. Verification analyses were carried out with samples derived from a minimum of 4 wild type and 4 dystrophic mice. Comparative tissue proteomics was carried out by standardized procedures, as previously described in detail [40,41]. Mice were sacrificed in the Bioresource Unit of the University of Bonn and muscle specimens were quick-frozen in liquid nitrogen and then transported on dry ice to Maynooth University [42]. Samples were stored at −80 °C prior to proteomic analysis. Muscle samples were homogenised in lysis buffer (4% SDS, 100 mM Tris-Cl, pH 7.6, 0.1 M dithiothreitol) using a handheld homogeniser from Kimble Chase (Rockwood, TN, USA), briefly treated in a sonicating water bath, and then heated for 3 min at 95 °C. Suspensions were centrifuged at 16,000× *g* for 5 min and the protein-containing supernatant extracted for subsequent analysis [43]. The Pierce 660 nm Protein Assay system was used to determine protein concentration [44]. EOM extracts were further processed for mass spectrometric analysis. Samples were mixed with 200 µL of 8 M urea, 0.1 M Tris pH 8.9 in filter units and centrifuged at 14,000× *g* for 15 min. For filter-aided sample preparation, processing was carried out according to the standardized FASP protocol [45].

### 2.3. Label-Free Liquid Chromatography Mass Spectrometry and Proteomic Data Analysis

The label-free liquid chromatography mass spectrometric analysis of EOMs from wild type versus *mdx-4cv* mice was carried out using a Thermo Orbitrap Fusion Tribrid mass spectrometer (Thermo Fisher Scientific, Waltham, MA, USA). Details of the proteomic workflow describing all preparative steps and analytical procedures using data-dependent acquisition, as well as bioinformatic data handling, were recently outlined in detail [41]. A Thermo UltiMate 3000 nano system was used for reverse-phased capillary high-pressure liquid chromatography and directly coupled in-line with the Thermo Orbitrap Fusion Tribrid mass spectrometer. The qualitative data analysis of mass spectrometric files was carried out with the help of the UniProtKB-SwissProt *Mus musculus* database with Proteome Discoverer 2.2 using Sequest HT (Thermo Fisher Scientific) and Percolator. For protein identification, the following crucial search parameters were employed: (i) a value of 0.02 Da for MS/MS mass tolerance, (ii) a value of 10 ppm for peptide mass tolerance, (iii) variable modification settings for methionine oxidation, (iv) fixed modification settings in relation to carbamido-methylation and (v) tolerance for the occurrence of up to two missed cleavages. Peptide probability was set to high confidence. A minimum XCorr score of 1.5 for 1, 2.0 for 2, 2.25 for 3 and 2.5 for 4 charge state was employed for the filtering of peptides [40]. The software analysis programme Progenesis QI for Proteomics (version 2.0; Nonlinear Dynamics, a Waters company, Newcastle upon Tyne, UK) was used to carry out quantitative label-free data analysis. Proteome Discoverer 2.2 using Sequest HT (Thermo Fisher Scientific) and a percolator were employed for the identification of peptides and proteins. Datasets were imported into Progenesis QI software for further analysis. Following the review of protein identifications, only those data that agreed with a crucial set of criteria were deemed as differentially expressed species between experimental groups based on statistical significance and high confidence. The criteria included an ANOVA *p*-value of ≤0.01 between experimental groups, and proteins with ≥2 unique peptides contributing to the identification. The Progenesis QI programme calculated the mean abundance for individual protein species in each experimental condition to determine the maximum fold change for particular proteins. Condition-vs-condition matrixes with mean values were then used to determine the maximum fold change between any two condition’s mean protein abundances [41]. The raw MS files generated by this proteomic study were deposited under the unique identifier ‘j4867’ to the Open Science Foundation (https://osf.io/j4867/ (accessed on 15 June 2021)). The standard bioinformatic analysis tools PANTHER [46] and STRING [47] were utilized for the identification of protein classes and for the characterisation of potential protein interaction patterns, respectively.

### 2.4. Comparative Immunoblot Analysis

For the independent evaluation of the differential expression levels of myosin isoform MyHC14 in wild type EOM versus wild type diaphragm, as identified by mass spectrometry, comparative immunoblotting was carried out under standard conditions [48]. Labelling of glyceraldehyde-3-phosphate dehydrogenase and fibronectin were used to evaluate concentration levels in *mdx-4cv* EOM as compared to wild type EOM preparations. EOM and diaphragm samples were prepared in Laemmli-type sample buffer and heated for 30 min at 37 °C. For gel electrophoresis and immunoblotting analysis, 20µg protein per lane were ran on Invitrogen Bolt 4–12% Bis-Tris gels. Coomassie staining of protein gels was performed with InstantBlue Coomassie Protein Stain [39]. For immunoblotting, gel electrophoretically separated proteins were transferred to nitrocellulose membranes, blocked in fat-free milk solution, and incubated in 1:1000 diluted primary antibody overnight. The subsequent detection with 1:1000 diluted peroxidase-conjugated secondary antibodies was carried out using the enhanced chemiluminescence method [40]. Statistical analysis of immunoblots was carried out using ImageJ software (NIH, Bethesda, MD, USA), along with Microsoft Excel, in which statistical significance was based on a *p*-value  ≤  0.05.

### 2.5. Immunofluorescence Microscopy

In order to evaluate the expression of glyceraldehyde-3-phosphate dehydrogenase in wild type versus *mdx-4cv* EOM muscle, immunofluorescence microscopy was carried out by standardized methodology in combination with histological staining [49]. Freshly dissected skeletal muscle specimens from mice were quick-frozen in liquid nitrogen-cooled isopentane and 10 μm sections were cut in a cryostat [50]. Tissue sections were fixed in a 1:1 (*v/v*) mixture of methanol and acetone for 10 min at room temperature and then blocked with 1:20 diluted normal goat serum for 30 min at room temperature. Primary antibodies to myosin heavy chain MyHC14 and glyceraldehyde-3-phosphate dehydrogenase were diluted 1:200 and 1:400, respectively, in carrageenan-containing and phosphate-buffered saline for overnight incubation at 4 °C. The buffer was made by mixing 100 mL phosphate-buffered saline with 0.7 g carrageenan and 10 mg sodium azide. Tissue specimens were carefully washed and then incubated with fluorescently labelled secondary antibodies, using 1:500 diluted anti-mouse RRX antibody for 45 min at room temperature [40]. Nuclei were counter-stained with 1 μg/mL bis-benzimide Hoechst 33,342. Antibody-labelled EOM sections were embedded in Fluoromount G medium and viewed under a Zeiss Axioskop 2 epifluorescence microscope equipped with a digital Zeiss AxioCam HRc camera (Carl Zeiss Jena GmbH, Jena, Germany).

## 3. Results and Discussion

In order to elucidate the unique cell biological and biochemical status of EOMs among other types of skeletal muscles [2], this study has focused on the refined proteomic analysis of this subtype of contractile tissue using an Orbitrap Fusion Tribrid mass spectrometer. Based on the mass spectrometric identification of the accessible EOM proteome, comparative analysis of wild type versus dystrophic *mdx-4cv* muscle preparations was carried out to investigate the underlying protein expression profile of the relatively mild phenotype of dystrophin-deficient EOMs in dystrophinopathy [19].

### 3.1. The Proteomic Profile of Extraocular Muscle

The analytical workflow used in this study is outlined in Figure 1. The mass spectrometry-based proteomic profiling of crude extracts from the EOM cone resulted in the identification of a large number of both muscle-specific marker proteins and core proteins of the sarcomere-associated contractile apparatus [51].

The multi-consensus file of unequivocally identified proteoforms was submitted to a public data repository (Open Science Foundation, OSF Facility, Frankfurt, Germany), under the unique identifier ‘j4867’ (https://osf.io/j4867/ (accessed on 15 June 2021)). The proteomic analysis presented in this report identified 2521 protein species in wild type EOM samples and 2331 protein species in *mdx-4cv* EOM samples. Table 1 lists the proteomic identification of sarcomeric proteins and related isoforms in EOM preparations from wild type mice. General marker proteins that exhibit the highest level of enriched expression in skeletal muscles according to the Human Protein Atlas [52] were identified as myosin heavy chain MyHC-IIa (MYH2), myosin-binding protein C (MYBPC1), the Z-disk component myotilin (MYOT), the M-band associated enzyme beta-enolase (ENO3), the half-sarcomere spanning giant protein titin (TTN), the actin-binding protein nebulin (NEB) of the thin filament and the skeletal muscle LIM-protein 1 (FHL1) (Table 1). Additional muscle-associated markers included the fast sarcoplasmic reticulum Ca^2+^-ATPase SERCA1 (Q8R429; Atp2a1 gene; 27.5% coverage; 19 unique peptides; 109.4 kDa), the slow sarcoplasmic reticulum Ca^2+^-ATPase SERCA2 (O55143; Atp2a2 gene; 16.4% coverage; 12 unique peptides; 114.8 kDa), the muscle-type glycolytic enzyme phosphofructokinase (P47857; Pfkm gene; 3.9% coverage; 2 unique peptides; 85.2 kDa) and the muscle-specific oxygen transporter myoglobin (P04247; Mb gene; 28.6% coverage; 5 unique peptides; 17.1 kDa) [39,53,54].

### 3.2. Proteomic Profile of the Sarcomere from Extraocular Muscle

An extensive list of sarcomeric proteins found in wild type EOM is listed in Table 1, including proteins of the thick myosin filament, the thin actin filament, the Z-disk, the M-band region, the titin filament and the auxiliary nebulin filament [55,56], as well as components of the sarcomere-attached cytoskeletal network [57]. The general arrangement of these components in the sarcomere is diagrammatically shown in Figure 2a. In total, this study established 15 different myosin heavy chains [17] to be associated with EOMs. This included the sarcomeric forms myosin-1 (MyHC-IId, muscle), myosin-2 (MyHC-IIa, muscle), myosin-3 (MyHC-embryonic), myosin-4 (MyHC-IIb, muscle), myosin-7 (MHyC-I, slow) and myosin-8 (MyHC-perinatal), as well as the specialized myosins named myosin-13, myosin-14, and myosin-15 [58,59].

The crucial myosin-binding protein present in EOMs was determined to be the slow-type MYBP-C1. The identified cytoskeletal myosins included the MyHC-cellular types A (myosin-9) and B (myosin-10). Besides smooth muscle myosin-11 heavy chain, 4 unconventional myosins were also identified by mass spectrometry, i.e., myosin-6 (E9Q175, Myo6, 144.7 kDa), myosin VC (E9Q1F5, Myo5c, 202.6 kDa), myosin XVIIIa (K3W4L0, Myo18a, 230.8 kDa) and myosin XVIIIb (E9PV66, Myo18b, 288.7 kDa). Myosin light chains included the muscle-specific isoforms MLC-1/3, MLC-2, and MLC-3 [36,60].

**Figure 2 life-11-00595-f002:**
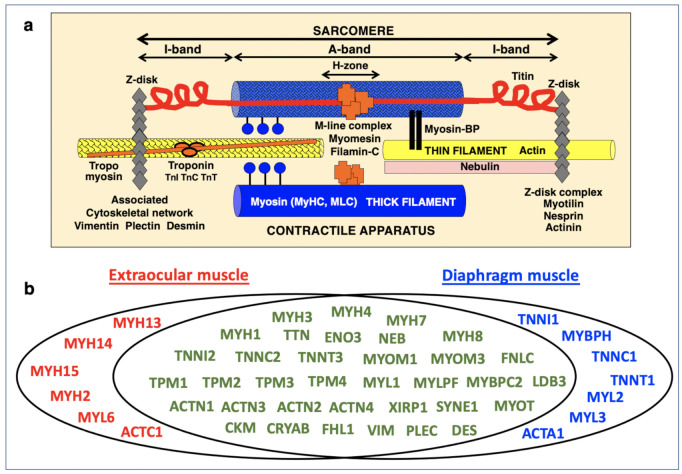
Comparative display of mass spectrometrically identified proteins of the sarcomere in extraocular muscle versus diaphragm from wild type mice. (**a**) Diagram of the main components of the contractile apparatus. (**b**) Venn diagram with the proteomic profile of the two types of investigated skeletal muscles.

Mass spectrometric analysis established major proteins of the thin filament in EOMs, including alpha-actin (ACTC1), nebulin, the tropomyosins TPM1 (alpha-1), TPM2 (beta), TPM3 (alpha-3) and TPM4 (alpha-4), and the troponin subunits TnC (TNNC2), TnI (TNNI2) and TnT (TNNT3) [59]. In agreement with a previous study [34], a large number of Z-disk-associated proteins were identified in EOMs, including the alpha-actinins ACTN1, ACTN2, ACTN3 and ACTN4, desmin, vimentin, plectin, alphaB-crystallin, LIM domain-binding protein LDB3, myotilin, nesprin-1 and xin actin-binding repeat-containing protein 1, as well as muscle-specific filamin-C [61]. Established markers of the M-band region in EOMs included muscle-type creatine kinase, myomesin-1, myomesin-3, adenylate kinase AK1, skeletal muscle LIM-protein FHL1 and muscle-specific beta-enolase [62].

### 3.3. Mass Spectrometric Identification of Proteins Specifically Expressed in Extraocular Muscle

In order to determine enriched expression levels of specific sarcomere-associated proteins in EOMs, the proteomic analysis of extracts from the wild type EOM cone was compared to the recently established proteome of wild type diaphragm muscle [39]. Figure 2b shows a Venn diagram of EOM versus diaphragm and illustrates the large cohort of overlapping expression of proteins belonging to the thick filament, the thin filament, the Z-disk, the M-band, auxiliary filaments and the sarcomere-associated cytoskeletal network [55,56,57]. Interestingly, elevated protein levels in EOMs included the myosin heavy chain isoforms myosin-2 (fast MyHC-IIa; MYH2) [51], myosin-13 (MyHC13; MYH13) [63], myosin-14 (MyHC14; MYH14) [58,59] and myosin-15 (MyHC15; MYH15) [58,59], as well as myosin light polypeptide 6 (MYL6; MLC-3 isoform) [60] and slow cardiac alpha-actin-1 (ACTC1) [64]. In contrast, diaphragm muscle contained apparently higher concentrations of myosin light chains MLC-2 (MYL2) and MLC-3 (MYL3), muscle alpha-actin-1 (ACTA1), the troponin isoforms TNNI1, TNNC1 and TNNT1 and myosin-binding protein MYBP-H [36,39]. These findings confirm that MyHC13 (myosin-13) represents an excellent marker of EOMs [17,34], and additionally establishes the two ancient myosins MyHC14 (myosin-14) and MyHC15 (myosin-15) as highly enriched components of this type of skeletal muscle [58,59]. Immunoblotting confirmed the differential expression pattern of myosin MyHC14 in wild type EOM versus diaphragm muscle. In contrast to comparable concentration levels of glyceraldehyde-3-phosphate dehydrogenase, the immunoblot analysis of MyHC14 demonstrated a significant elevation of this myosin isoform in EOMs as compared to diaphragm muscle (Figure 3).

The EOM-specific expression and distribution of specialized types of myosin heavy chains, such as MyHC13, MyHC14 and MyHC15 [17], which surround the globe structure within the ocular cavity, could be related to the unusual physiological properties of this type of muscle. In addition, the broad isoform expression pattern in EOMs including both slow and fast myosin isoforms, i.e., slow MyHC-I, fast MyHC-IIa, fast MyHC-IIb and fast MyHC-IId, as well as embryonic MyHC3 and perinatal MyHC8, form probably the functional basis for an extremely wide range of possible eye movements. This allows complex movements of the eyeball in relation to its horizontal and vertical axis ranging from slow vergence to rapid saccades [6]. EOMs are capable of both considerable eccentric contraction patterns and fibre twitching at high frequency without tetanus, and this is at least partially provided by the kinetic properties of the unique combination of myosins and their distribution in EOMs. For example, the embryonic MyHC3 isoform is located at the terminal region of contractile fibres, while the super-fast MyHC13 isoform is positioned at the central endplate [63].

### 3.4. Comparative Proteomic Profiling of Extraocular Muscle from the Dystrophic mdx-4cv Model of Duchenne Muscular Dystrophy

Dystrophinopathy is the most frequently inherited neuromuscular disease of early childhood and is characterized by progressive skeletal muscle degeneration [21,26], in combination with reactive myofibrosis [28] and sterile inflammation [40]. Genetic rearrangements in the DMD gene cause the almost complete loss of the full-length dystrophin isoform Dp427-M [21] and the simultaneous disintegration of the dystrophin-glycoprotein complex [22]. Established genetic animal models of dystrophinopathy, such as the mdx-4cv mouse [65,66,67], reflect many of the multifaceted and body-wide alterations seen in Duchenne patients, including necrosis, fibrosis and inflammation in the diaphragm muscle [39,68], cardiomyopathic changes [69] and neuronal deficiencies [70], as well as secondary abnormalities in the liver [71], kidney [48], and spleen [40]. The molecular and cellular pathogenesis of muscular dystrophy is also mirrored by characteristic protein changes in biofluids, such as mdx-4cv serum, urine and saliva [72,73,74]. Building on the well-established muscle degeneration and accompanying effects on multiple other organ systems in the mdx-4cv mouse model, it was of interest to determine the pathobiochemical signature of apparently spared EOMs [75].

Detailed histological, cell biological and biochemical studies of dystrophin-deficient EOM have been carried out [76,77,78,79,80,81] and are supplemented here with findings on proteome-wide changes. Table 2 lists significantly altered proteins in *mdx-4cv* EOM. This includes characteristic increases in the glycolytic enzymes glyceraldehyde-3-phosphate dehydrogenase, enolase, and lactate dehydrogenase [82], as well as the molecular chaperones heat shock protein 1 beta (HspB1) and heat shock cognate 71 kDa protein (HspA8) [83]. The findings from previous immunoblotting and mass spectrometric surveys of EOMs that were extracted from the spontaneously mutated *mdx-23* mouse model of dystrophinopathy agree with these changes in distinct protein families [37,38]. In contrast to comparable expression levels of the dystrophin-associated glycoprotein beta-dystroglycan, higher levels of the molecular chaperones alphaB-crystallin, cvHsp/HspB7, Hsp25/HspB1 and Hsp90 were previously demonstrated to exist in dystrophin-deficient EOMs. The apparent up-regulation of heat shock proteins in both *mdx* and *mdx-4cv* EOMs is an indication of a robust cellular stress response in dystrophin-deficient EOMs [84]. Changes in HspB1 might be suitable to establish this small heat shock protein as a marker of the stress response in EOMs in the dystrophic phenotype [37].

The relatively mild phenotype of EOMs in X-linked muscular dystrophy [19,20] is probably closely related to the special biochemical and physiological features of the muscles surrounding the eyeball, such as the longitudinal distribution of neuromuscular junctions [9,10,11], the considerable capacity for fibre regeneration [13,14], exceptional fatigue resistance even in fast-twitching fibre populations [16], and an extensive calcium extrusion system [15]. In most skeletal muscles, dystrophin deficiency causes a collapse of sarcolemmal integrity and a concomitant increase in micro-rupturing of the surface membrane, which triggers influx of Ca^2+^-ions into myofibres and associated enhanced Ca^2+^-dependent proteolytic activity in the sarcosol [25,27]. The efficient and swift removal of cytosolic calcium from EOM fibres and the enhanced ion buffering capacity of EOMs might therefore play a key role in the protection from dystrophic changes [18,76,79]. Another important factor might be the relatively low concentration of dystrophin isoform Dp427-M in EOMs. The sub-sarcolemmal dystrophin lattice might not play the same crucial role in the membrane cytoskeleton and linkage of the intracellular actin filaments to extracellular laminin via the dystrophin-glycoprotein complex in EOMs as compared to other skeletal muscles [36]. Importantly, previous studies established an up-regulation of the autosomal dystrophin homologue named utrophin Up-395 and associated rescue of sarcolemmal glycoproteins such as beta-dystroglycan in dystrophin-deficient EOMs [37,77,78]. Thus, full-length utrophin might substitute for dystrophin in EOMs and thereby stabilize its trans-sarcolemmal cytolinker function and prevent secondary damage to myofibres.

**Table 2 life-11-00595-t002:** Comparative proteomic analysis of the *mdx-4cv* extraocular muscle cone.

Accession	Protein	Gene	Peptides	Annova (*p*)	Fold Change
P16858	Glyceraldehyde-3-phosphate dehydrogenase	GAPDH	14	0.01427	+18.45
P14602	Heat shock protein 1, beta	HSPB1	25	0.04444	+15.62
P63017	Heat shock cognate 71 kDa protein	HSPA8	24	0.00894	+10.22
G3XA25	Acetyl-CoA acetyltransferase 2	ACAT2	24	0.04001	+7.56
P06151	Lactate dehydrogenase 1, A chain	LDHA	14	0.02856	+5.93
P19536	Hydroxyacyl-coenzyme A dehydrogenase	HADH	22	0.04369	+3.38
P17182	Alpha-enolase	ENO1	19	0.04825	+3.13
P47740	Aldehyde dehydrogenase family 3 member A2	ALDH3A2	15	0.04301	+2.94
E9QNH7	Acyl-CoA-binding domain-containing protein 5	ACBD5	6	0.03134	+2.43
P07724	Albumin	ALB	45	0.03520	+1.76
P19096	Fatty acid synthase	FASN	119	0.02425	+1.69
G5E8R1	Tropomyosin alpha-1 chain	TPM1	11	0.03985	−1.98
Q9DB20	ATP synthase subunit O, mitochondrial	ATP5PO	7	0.03449	−2.07
P37040	NADPH-cytochrome P450 reductase	POR	12	0.04813	−2.45
Q9R0P5	Destrin	DSTN	9	0.03484	−2.66
Q9D3D9	ATP synthase subunit delta, mitochondrial	ATP5F1D	4	0.04228	−2.71
Q9CRB9	MICOS complex subunit Mic19	CHCHD3	11	0.04353	−2.90
Q9DBC7	cAMP-dependent protein kinase type I-alpha	PRKAR1A	2	0.02217	−2.91

The bioinformatic analysis of the proteomic survey of mdx-4cv EOMs is summarized in Figure 4, which displays the PANTHER analysis of the overall protein profile [46], the heat map of proteomic changes and the findings from the STRING analysis of potentially altered protein interaction hubs [47]. The overall distribution of protein families was shown not to be majorly different between wild type and dystrophic mdx-4cv EOM and agrees with previous mass spectrometric surveys [37,38]. The heat map illustrates the distribution pattern of changes between normal and dystrophic specimens. In contrast to other sub-types of dystrophin-deficient skeletal muscles [31,39,42,84,85,86,87], Dp427-lacking EOMs seem to exhibit relatively minor proteome-wide changes.

The STRING interaction network indicates that the drastically increased enzyme glyceraldehyde-3-phosphate dehydrogenase is positioned centrally within an altered protein hub in dystrophic EOMs. Immunofluorescence microscopy and immunoblotting was employed to compare the expression of glyceraldehyde-3-phosphate dehydrogenase. As illustrated in Figure 5, immunofluorescence labelling of this glycolytic enzyme shows elevated levels in dystrophin-deficient EOMs, as compared to relatively comparable levels of myosin MyHC14. Immunoblotting also indicates increased levels of glyceraldehyde-3-phosphate dehydrogenase in mdx-4cv EOMs, as compared to comparable expression of fibronectin. An approximately 2-fold increase in the expression of glyceraldehyde-3-phosphate dehydrogenase has previously been identified in dystrophic mdx-23 diaphragm preparations [84]. Thus, a potential shift to more glycolytic metabolism appears to be associated with the dystrophic phenotype, and this seems to be especially striking in mdx-4cv EOMs, suggesting these types of metabolic enzymes as biomarker candidates for studying dystrophinopathy-related changes in bioenergetic pathways. The comparable levels of fibronectin in wild type versus dystrophin-deficient EOMs indicates the lack of reactive myofibrosis in this type of muscle, which is otherwise seen in most contractile tissues affected in the dystrophic phenotype [28,33,68].

In addition, elevated expression levels were shown for cytosolic acetyl-CoA acetyltransferase ACAT2, aldehyde dehydrogenase family 3 member A2 and acyl-CoA-binding domain-containing protein 5, albumin, hydroxyacyl-coenzyme A dehydrogenase and fatty acid synthase. The proteomic changes in key enzymes of the glycolytic pathway and anaerobic metabolism suggest a potential metabolic shift in mdx-4cv EOMs [88]. A concomitant decrease was observed in certain mitochondrial proteins including the ATP synthase subunit O, NADPH-cytochrome P450 reductase, ATP synthase subunit delta and MICOS complex subunit Mic19. Decreased proteins also included Tropomyosin alpha-1 chain, the actin-depolymerizing protein destrin and cAMP-dependent protein kinase type I-alpha.

A potential weakness and bioanalytical limitations of this report relate to the usage of an animal model of Duchenne muscular dystrophy instead of patient samples, and the focus on changed protein concentration using peptide mass spectrometry. With the exception of the diaphragm, the mdx-4cv mouse exhibits relatively mild symptoms of fibre wasting in its general musculature. Thus, the findings from this study should ideally be extended to characterize muscle specimens from Duchenne patients, which is, however, extremely difficult in the case of EOMs. Proteomics focuses on the mass spectrometric identification of individual proteoforms and can be routinely employed for comparative studies. However, it is important to realize that the establishment of abundance changes in individual proteins does not provide detailed information on the underlying regulatory mechanisms. It will therefore be important to supplement the new proteomic datasets with future analyses of the biochemical, physiological and cell biological properties of dystrophin-deficient EOMs.

## 4. Conclusions

The systematic mass spectrometry-based proteomic survey of EOM specimens established distinct myosin isoforms of the MyHC category as new sarcomeric marker candidates of this specialized type of skeletal muscle, i.e., MyHC14 and MyHC15, besides confirming MyHC13 as an EOM enriched component. The drastically elevated levels of MyHC14 in wild type EOMs as compared to wild type diaphragm muscle were clearly confirmed by immunoblotting. This makes MyHC14 a suitable biomarker of EOMs and it remains to be elucidated what exact physiological role this particular proteoform of the myosin complex plays in the contractile kinetics and functional adaptability of EOMs. Comparative proteomics of wild type versus dystrophic specimens indicates that an apparent metabolic shift from oxidative metabolism to a more glycolytic pathway and heightened cellular stress response exists in *mdx-4cv* EOM. Elevated levels of the glycolytic enzyme glyceraldehyde-3-phosphate dehydrogenase appear to be associated with dystrophic alterations in mildly affected EOMs. These alterations in the cellular homeostasis may serve as mechanisms to compensate for deficits induced by the dystrophin loss in this model of Duchenne muscular dystrophy.

## Figures and Tables

**Figure 1 life-11-00595-f001:**
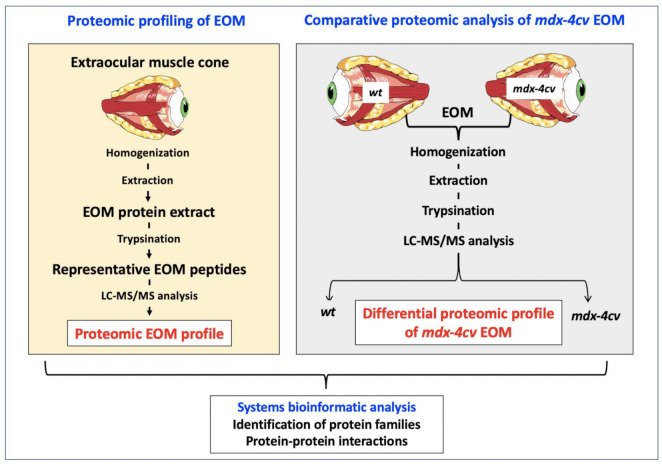
Overview of the proteomic profiling approach to characterize extraocular muscle (EOM), as well as determine changes in the *mdx-4cv* mouse model of Duchenne muscular dystrophy.

**Figure 3 life-11-00595-f003:**
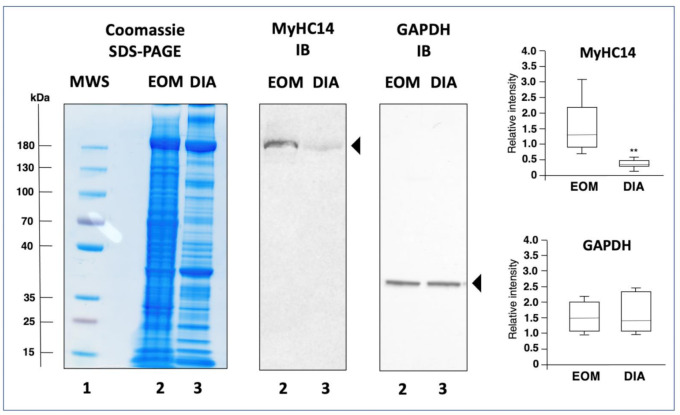
Immunoblot analysis of extraocular muscle (EOM) versus diaphragm (DIA) muscle from wild type mice. Shown is an InstantBlue Coomassie stained protein gel using sodium dodecyl sulphate polyacrylamide gel electrophoresis (SDS-PAGE) with molecular weight standards (lane 1; MWS), wild type EOM (lane 2) and wild type DIA (lane 3) samples, as well as identical nitrocellulose replicas used for immunoblotting (IB) and labelled with antibodies to myosin heavy chain MyHC14 and glyceraldehyde-3-phosphate dehydrogenase (GAPDH). In the adjacent panels are shown the statistical analysis of immunoblotting (Mann–Whitney U test; *n* = 8; ** *p* < 0.01). The value of molecular mass standards (×10^−3^ kDa) is marked on the left side of the gel.

**Figure 4 life-11-00595-f004:**
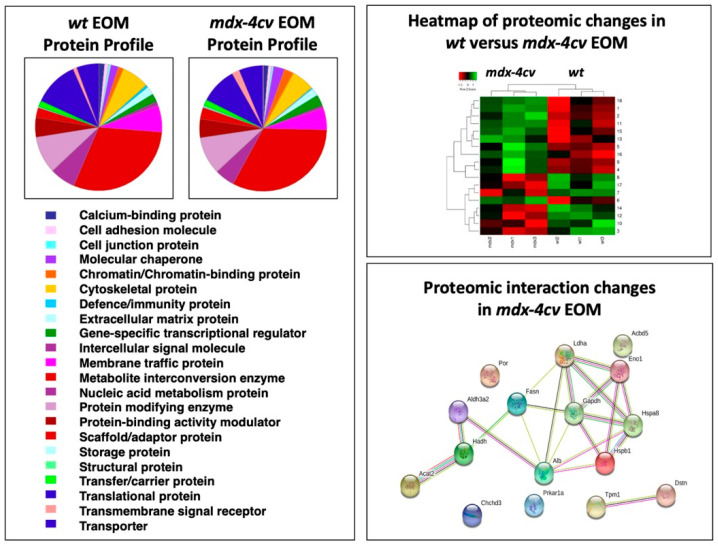
Bioinformatic analysis of the comparative proteomic profiling of extraocular muscle (EOM) from wild type (*wt*) versus the *mdx-4cv* model of X-linked muscular dystrophy. Shown is the result of the bioinformatic PANTHER analysis [46] of the distribution of protein classes within the EOM proteome from normal versus dystrophic mice. In addition, the heat map of the comparative proteomic analysis of *wt* versus *mdx-4cv* EOM is displayed, which shows the findings from hierarchical clustering of the mean protein expression values of statistically significant differentially abundant EOM proteins. Potential changes in protein–protein interaction patterns in *mdx-4cv* EOMs were determined with the help of the bioinformatics software programme STRING [47].

**Figure 5 life-11-00595-f005:**
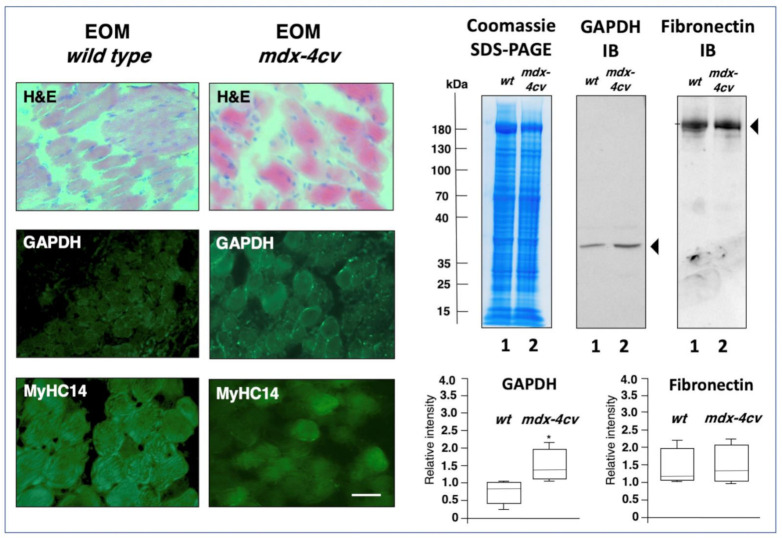
Histological and immunofluorescence microscopical characterization, as well as immunoblot analysis, of extraocular muscle (EOM) from the *mdx-4cv* mouse model of Duchenne muscular dystrophy. Shown are transverse cryosections of wild type (*wt*) and *mdx-4cv* EOMs stained with haematoxylin and eosin (H&E) and labelled with antibodies to glyceraldehyde-3-phosphate dehydrogenase (GAPDH) and myosin heavy chain MyHC14. Bar equals 30 μm. Shown is an InstantBlue Coomassie (CBB) stained protein gel with wild type EOM (lane 1) and *mdx-4cv* EOM (lane 2) samples, as well as identical nitrocellulose replicas used for immunoblotting and labelled with antibodies to glyceraldehyde-3-phosphate dehydrogenase (GAPDH) and fibronectin (FN). In the below panels are shown the statistical analysis of immunoblotting (Mann–Whitney U test; *n* = 4; * *p* < 0.05). The value of molecular mass standards (x10^−3^ kDa) is marked on the left side of the gel.

**Table 1 life-11-00595-t001:** Proteomic identification of sarcomere-associated proteins and related isoforms in mouse extraocular muscle.

Accession	Protein	Gene	Score	Coverage %	Peptides	Molecular Mass (kDa)
E9Q8K5	Titin, muscle- specific	TTN	171.5	2.14	52	3713.7
Q5SX40	Myosin-1 heavy chain, MyHC-IId, fast muscle	MYH1	459.6	31.10	63	223.2
G3UW82	Myosin-2 heavy chain, MyHC-IIa, fast muscle	MYH2	382.7	31.15	60	223.1
P13541	Myosin-3 heavy chain, MyHC-embryonic, muscle	MYH3	125.1	11.70	24	223.7
Q5SX39	Myosin-4, MyHC-IIb, fast muscle	MYH4	472.9	32.75	68	222.7
Q91Z83	Myosin-7 heavy chain, MyHC-I, slow muscle	MYH7	96.04	10.70	20	222.7
P13542	Myosin-8 heavy chain, MyHC-perinatal, muscle	MYH8	277.9	24.68	47	222.6
Q8VDD5	Myosin-9 heavy chain, MyHC-cellular, type A	MYH9	87.81	10.61	20	226.2
Q61879	Myosin-10 heavy chain, MyHC-cellular, type B	MYH10	20.79	2.68	5	228.8
A0A2R8VHF9	Myosin-11 heavy chain, smooth muscle	MYH11	255.1	20.07	44	223.2
B1AR69	Myosin-13 heavy chain, extraocular muscle	MYH13	217.41	17.03	31	223.4
K3W4R2	Myosin-14 heavy chain, MyHC-eom, developmental	MYH14	25.40	3.45	6	228.4
E9Q264	Myosin-15 heavy chain, extraocular muscle	MYH15	10.20	1.56	3	221.7
P05977	Myosin light chain MLC-1/3, muscle	MYL1	74.36	45.21	8	20.6
A0A0U1RP93	Myosin light chain MLC-2, muscle	MYLPF	28.02	10.07	1	16.9
Q60605	Myosin light chain MLC-3, muscle	MYL6	56.91	54.30	7	16.9
D3YU50	Myosin-binding protein C, slow	MYBPC1	20.96	5.24	4	126.5
A0A571BF46	Nebulin	NEB	15.55	1.32	8	866.5
P68033	Actin, alpha, skeletal muscle	ACTC1	382.05	66.84	26	42.0
P60710	Actin, beta, cytoplasmic	ACTB	384.11	52.53	25	41.7
E9Q452	Tropomyosin alpha-1	TPM1	119.92	40.93	14	32.5
A2AIM4	Tropomyosin beta	TPM2	112.31	41.55	16	33.0
D3Z6I8	Tropomyosin alpha-3	TPM3	47.89	43.72	12	28.7
A0A571BEU1	Tropomyosin alpha-4	TPM4	4.33	14.56	3	18.4
P20801	Troponin C, muscle	TNNC2	3.99	11.88	2	18.1
A0A1B0GRY8	Troponin I, fast muscle	TNNI2	23.66	13.95	2	20.2
A2A6I8	Troponin T, fast muscle	TNNT3	8.66	20.08	4	28.3
A1BN54	Alpha-actinin-1	ACTN1	36.58	8.34	6	102.7
Q9JI91	Alpha-actinin-2	ACTN2	13.09	7.83	6	103.8
O88990	Alpha-actinin-3	ACTN3	11.30	5.11	4	103.0
P57780	Alpha-actinin-4	ACTN4	65.92	20.39	14	104.9
P31001	Desmin	DES	27.89	15.99	8	53.5
P20152	Vimentin	VIM	72.37	24.89	8	53.7
E9Q3W4	Plectin	PLEC	22.89	3.17	12	498.8
P23927	Alpha-B-crystallin	CRYAB	14.36	36.57	7	20.1
E9PYJ9	LIM domain-binding protein 3	LDB3	10.34	3.98	2	72.3
Q9JIF9	Myotilin	MYOT	10.23	6.45	3	55.3
A0A1L1STC6	Nesprin-1	SYNE1	4.29	0.31	2	1009.3
A0A571BDS0	Xin actin-binding repeat-containing protein 1	XIRP1	2.03	1.05	2	196.6
Q8VHX6	Filamin-C, muscle-specific	FNLC	21.04	2.53	6	290.9
Z4YJF5	Myomesin-1	MYOM1	2.09	0.83	1	175.4
A2ABU4	Myomesin-3, slow, extraocular muscle	MYOM3	5.42	2.08	2	161.7
P07310	Creatine kinase, muscle-type	CKM	48.98	29.92	11	43.0
Q9R0Y5	Adenylate kinase, AK1	AK1	17.30	16.49	2	21.5
P97447	Skeletal muscle LIM-protein 1	FHL1	10.47	5.89	2	61.8
P21550	Beta-enolase, muscle-specific	ENO3	152.38	42.63	18	47.0

## Data Availability

The raw MS files generated by this proteomic study have been deposited under the unique identifier ‘j4867’ to the Open Science Foundation (https://osf.io/j4867/).

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
