# Peer review of "Mass Spectrometric Profiling of Extraocular Muscle and Proteomic Adaptations in the mdx-4cv Model of Duchenne Muscular Dystrophy"

_life, 2021, doi:10.3390/life11070595_

Round 1

Reviewer 1 Report

line 109:  muscle phenotype of mdx mice shows different phases which do not match with the age of mice that have been used for the experiements. At 3-6 months mdx mice there is a phase of muscle necrosis. On the other hand, muscle wasting, heart failure....appears 15 months.  What is the rationale for using  12 months old mice to perform proteomics of EOM?

line 109: the analysis was peformed on each  EO muscle subtype or the six EO muscles were pooled and then processed for label-free mass spec?

lines 112-113: it is not clear how many biological samples and how many replicates of each biological samples have been used for statistical analysis

line 121: proteomic analysis has been carried out with proteins enriched in the 16000 g pellet:  1) Why?; 2) Which subcellular fractions are present in the 16000 g pellet

line 128: label free mass-spec has much lower precison compared to TMT mass spec. I was wondering whether  or not the specimens where fractionated before injection. How many proteins were in total identified?

line 261: can something more be said about levels of these proteins in EOM vs diaphragm

line 279 :  not clear the meaning of  " ...in different sub-types of skeletal muscles": fast? Slow ? diaphragm?

line 317: Table 2 does not show  expression of utrophin and of proteins of the dystroglycan complex. Why?

line 338: immunofluorescence is a poor quantitative method. I would not be too much confident on quantitative assessment based on immunofluorescence

Author Response

Reviewer 1, Comment 1: ‘line 109:  muscle phenotype of mdx mice shows different phases which do not match with the age of mice that have been used for the experiements. At 3-6 months mdx mice there is a phase of muscle necrosis. On the other hand, muscle wasting, heart failure....appears 15 months.  What is the rationale for using  12 months old mice to perform proteomics of EOM?’.

Response: We’d like to thank Reviewer 1 for pointing out the timeline of the pathogenesis of the mdx mouse. In our mass spectrometric analysis, we employed the mdx-4cv model of X-linked muscular dystrophy and used 12-month old animals in order to be able to compare the proteomic findings to our previous analysis of 12-month old dystrophic and highly fibrotic mdx-4cv diaphragm muscle (Murphy et al., 2019). Since EOMs are only mildly affected by deficiency in dystrophin, it was more important for us to produce proteomic data that could be compared to the already established diaphragm results than following the various phases of the pathogenesis of the mdx-4cv phenotype.
Reference [39]: Murphy, S.; Zweyer, M.; Raucamp, M.; Henry, M.; Meleady, P.; Swandulla, D.; Ohlendieck, K. Proteomic profiling of the mouse diaphragm and refined mass spectrometric analysis of the dystrophic phenotype. J. Muscle Res. Cell. Motil. 2019, 40, 9-28, doi: 10.1007/s10974-019-09507-z

Reviewer 1, Comment 2: ‘line 109: the analysis was peformed on each  EO muscle subtype or the six EO muscles were pooled and then processed for label-free mass spec?’.

Response: Due to the restricted amounts of EOM tissue being available from mice, we pooled the EOMs for the proteomic analysis outlined in this report. We actually used the entire EOM cone for the isolation of the combined EOM proteome. This point was also brought up by Reviewer 2 (Comment 3). We now outline this in more detail in the revised Methods section.
Revised Methods (line 116-119): ‘The eyeball and its surrounding tissues were carefully removed from the ocular cavity by bulbar exenteration. The entire EOM cone was then dissected out and extracted for the isolation of the combined EOM proteome’.

Reviewer 1, Comment 3: ‘lines 112-113: it is not clear how many biological samples and how many replicates of each biological samples have been used for statistical analysis’.

Response: The number of analysed specimens has now been listed more clearly in the revised methods section.
Revised Methods (lines 119-122): … The establishment of multi-consensus files was carried out with muscle samples from 6 wild type and 6 dystrophic mice. Comparative proteomics was performed with specimens from 3 wild type versus 3 dystrophic mice. Verification analyses were carried out with samples derived from a minimum of 4 wild type and 4 dystrophic mice’.

Reviewer 1, Comment 4: ‘line 121: proteomic analysis has been carried out with proteins enriched in the 16000 g pellet:  1) Why?; 2) Which subcellular fractions are present in the 16000 g pellet’.

Response: We’d like to thank Reviewer 1 for bringing up this point. There appears to be a misunderstanding due to wording issues in the Methods section. Our proteomic analysis was carried out with SDS-solubilized proteins in combination with the FASP method. Muscle samples were homogenised in SDS-containing lysis buffer, followed by sonication, a brief heating step and then centrifugation to remove any non-solubilized material and cellar debris.  Subsequently, the supernatant with the protein-containing fraction was used for proteomic analysis. The revised methods section has been re-worded accordingly.
Revised methods section (lines 130-131): ‘Suspensions were centrifuged at 16,000xg for 5 min and the protein-containing supernatant extracted for subsequent analysis [43]’.

Reviewer 1, Comment 5: ‘line 128: label free mass-spec has much lower precison compared to TMT mass spec. I was wondering whether  or not the specimens where fractionated before injection. How many proteins were in total identified?’.

Response: Our proteomic survey was carried out without subcellular fractionation procedures. The mass spectrometric analysis was performed on the protein supernatant following solubilization with SDS detergent. The combined data in multi-consensus files identified 2521 proteins in wild type EOM samples and 2331 proteins in mdx-4cv EOM samples. This information has now been added to the revised results section.
Revised Results section (lines 224-225): ‘… under the unique identifier ‘j4867’ (https://osf.io/j4867/). The proteomic analysis presented in this report identified 2521 protein species in wild type EOM samples and 2331 protein species in mdx-4cv EOM samples. Table 1 lists the proteomic …’.

Reviewer 1, Comment 6: ‘line 261: can something more be said about levels of these proteins in EOM vs diaphragm’.

Response: Section 3.3. outlines the mass spectrometric identification of proteins specifically expressed in extraocular muscle, as compared to findings from a previous proteomic survey of diaphragm muscle (Murphy et al., 2019). Although both analyses were carried out with comparable approaches on the same mass spectrometer, a Thermo Orbitrap Fusion Tribrid apparatus housed in the National Institute for Cellular Biotechnology at Dublin City University, we did not compare the proteomes of both muscle sub-types at the same time using the same experimental set up for a direct comparison of EOMs versus diaphragm. Thus, no conclusions about precise differences in abundance levels can be directly deduced from our current analysis of EOMs.

Reviewer 1, Comment 7: ‘line 279 :  not clear the meaning of  " ...in different sub-types of skeletal muscles": fast? Slow ? diaphragm?’.

Response: To avoid confusion, this sentence was changed on line 294 as follows: ‘Immunoblotting confirmed the differential expression pattern of myosin MyHC-14 in EOM versus diaphragm muscle’.

Reviewer 1, Comment 8: ‘line 317: Table 2 does not show  expression of utrophin and of proteins of the dystroglycan complex. Why?’.

Response: Due to the relatively small amounts of EOM tissue that can be extracted from mice, we carried out the proteomic analysis in this report using total protein extracts. Proteins belonging to the dystrophin-glycoprotein complex, such as dystroglycan, dystrobrevin and syntrophin were only identified by 1 unique peptide in wild type EOM preparations. Utrophin was only recognized by a utrophin fragment and 1 unique peptide in wild type EOMs. Thus, the detection of these relatively low-abundance proteins in EOMs appears to be below the level of detection of the mass spectrometric instrument used in this study.
However, our laboratory has identified all core members of the dystrophin complex in previous studies of other types of skeletal muscles, such as leg muscle or the diaphragm. Extensive subcellular fractionation procedures to prepare the microsomal fraction or highly purified sarcolemma vesicles usually enable the identification of dystrophin and associated glycoproteins by mass spectrometry.
For representative references from our laboratories, please see: 
Murphy, S.; Zweyer, M.; Henry, M.; Meleady, P.; Mundegar, R.R.; Swandulla, D.; Ohlendieck, K. Proteomic analysis of the sarcolemma-enriched fraction from dystrophic mdx-4cv skeletal muscle. J. Proteomics 2019, 191, 212-227, doi: 10.1016/j.jprot.2018.01.015
Murphy, S.; Zweyer, M.; Mundegar, R.R.; Henry, M.; Meleady, P.; Swandulla, D.; Ohlendieck, K. Concurrent Label-Free Mass Spectrometric Analysis of Dystrophin Isoform Dp427 and the Myofibrosis Marker Collagen in Crude Extracts from mdx-4cv Skeletal Muscles. Proteomes 2015, 3, 298-327, doi: 10.3390/proteomes3030298
We have also previously carried out an immunoblot analysis of beta-dystroglycan in EOMs and found comparable levels between wild type and mdx-23 mice (Lewis, C.; Ohlendieck, K. Proteomic profiling of naturally protected extraocular muscles from the dystrophin-deficient mdx mouse. Biochem. Biophys. Res. Commun. 2010, 396, 1024-1029, doi: 10.1016/j.bbrc.2010.05.052), so this protein would not be expected to be included in proteins with a changed abundance in dystrophin-deficient EOMs.

Reviewer 1, Comment 9: ‘line 338: immunofluorescence is a poor quantitative method. I would not be too much confident on quantitative assessment based on immunofluorescence’.

Response: While we basically agree with the opinion of Reviewer 1 that immunofluorescence microscopy does not represent a good method for quantitative assessments, this standard cell biological technique is nevertheless widely used as an illustrative approach to study the subcellular localization of proteins. Since the quality of the immunofluorescence images was questioned by Reviewer 2 (Comment 5), we have improved the cell biological images in Figure 5 and have also added the results of the immunoblotting of GAPDH in wild type versus mdx-4cv EOM preparations.

Reviewer 2 Report

“Mass spectrometric profiling of extraocular muscle and proteomic adaptations in the mdx-4cv model of Duchenne muscular dystrophy” is a very interesting and well performed study. I have the following comments, of which the point on the muscles used is crucial:

Introduction:

Lines 52-59: The information presented here is not absolutely necessary for the understanding of the paper but I agree that it is useful to mention the unique properties of the extraocular muscles. I suggest that you add the recent report of an additional type of neuromuscular junctions (Investigative Ophthalmology and Visual Science 2018; 59:539-) and that the text of point iv) is omitted as it is very controversial whether there is functional compartmentalization and it is not relevant to the present study.

Line 73: the list of references is very long and it is questionable that for ex refs 28-31 are all necessary.

Material and Methods and Results:

Line 108-: There is no description of how the extraocular muscles were dissected out and in the Results section the expression “the EOM cone” is used. In rodents the cone of muscles surrounding the eye also includes M retractor bulbi, which is not an extraocular muscle; it has characteristics of an ordinary striated muscle. Please check whether the retractor bulbi was included or not. It could explain why you did not find stronger differences between the muscle groups in your study. This needs to be clarified and is a very important, really crucial point!

Results and Discussion.

A comparison with previous studies needs to be added along with a discussion of the weaknesses of the study.

Figure 5

The quality of the top plates from EOM mdx-4cv in Eosin and MyHC14 needs to be significantly improved, even that of plate wt in Eosin.

Author Response

Reviewer 2, Comment 1: “Mass spectrometric profiling of extraocular muscle and proteomic adaptations in the mdx-4cv model of Duchenne muscular dystrophy” is a very interesting and well performed study. I have the following comments, of which the point on the muscles used is crucial: Introduction: Lines 52-59: The information presented here is not absolutely necessary for the understanding of the paper but I agree that it is useful to mention the unique properties of the extraocular muscles. I suggest that you add the recent report of an additional type of neuromuscular junctions (Investigative Ophthalmology and Visual Science 2018; 59:539-) and that the text of point iv) is omitted as it is very controversial whether there is functional compartmentalization and it is not relevant to the present study.

Response: We agree and have removed point (iv) and associated reference; and have instead added the suggested reference on neuromuscular junctions, now listed as new reference [11].
Revised lines 52-53: ‘…(ii) longitudinal dispersion of multiterminal neuromuscular junctions [9-11], …’.
New Reference [11]: Liu JX, Domellöf FP. A Novel Type of Multiterminal Motor Endplate in Human Extraocular Muscles. Invest Ophthalmol Vis Sci. 2018;59(1):539-548. doi: 10.1167/iovs.17-22554

Reviewer 2, Comment 2: ‘Line 73: the list of references is very long and it is questionable that for ex refs 28-31 are all necessary’.
Response: We agree and have reused previously quoted References 24 and 25 in revised line 77 and removed original References 28, 30 and 31 in order to reduce the overall number of references. Subsequent references have been re-numbered accordingly.

Reviewer 2, Comment 3: ‘Material and Methods and Results: Line 108-: There is no description of how the extraocular muscles were dissected out and in the Results section the expression “the EOM cone” is used. In rodents the cone of muscles surrounding the eye also includes M retractor bulbi, which is not an extraocular muscle; it has characteristics of an ordinary striated muscle. Please check whether the retractor bulbi was included or not. It could explain why you did not find stronger differences between the muscle groups in your study. This needs to be clarified and is a very important, really crucial point!
Response: The entire cone structure was dissected and analysed, which is now better described in the revised Methods section as follows (line 116-119): ‘… The eyeball and its surrounding tissues were carefully removed from the ocular cavity by orbital exenteration. The entire EOM cone was then dissected out and extracted for the isolation of the combined EOM proteome. …’.

Reviewer 2, Comment 4: ‘Results and Discussion. A comparison with previous studies needs to be added along with a discussion of the weaknesses of the study.
Response:  We agree and have added additional comparisons to previous studies in the revised discussion section, as well as now mention potential limitations of the study in relation to both proteomics and the animal model of dystrophinopathy.
Revised lines 345-351:  ‘… as well as the molecular chaperones heat shock protein 1 beta (HspB1) and heat shock cognate 71 kDa protein (HspA8) [82]. Findings from previous immunoblotting and mass spectrometric surveys of EOMs that were extracted from the spontaneously mutated mdx-23 mouse model of dystrophinopathy agree with these changes in distinct protein families [37,38]. In contrast to comparable expression levels of the dystrophin-associated glycoprotein beta-dystroglycan, higher levels of the molecular chaperones alphaB-crystallin, cvHsp/HspB7, Hsp25/HspB1 and Hsp90 were previously demonstrated to exist in dystrophin-deficient EOMs. The apparent up-regulation of heat shock proteins in both mdx and mdx-4cv EOM is an indication of a robust cellular stress response in dystrophin-deficient EOMs [83]. Changes in HspB1 might be suitable to establish this small heat shock protein as a marker of the stress response in EOMs in the dystrophic phenotype [37]’.
Revised lines 403-411: ‘An approximately 2-fold increase in the expression of glyceraldehyde-3-phosphate dehydrogenase has also been identified in dystrophic mdx-23 diaphragm preparations [86]. Thus, a potential shift to more glycolytic metabolism appears to be associated with the dystrophic phenotype, and this seems to be especially striking in mdx-4cv EOMs, suggesting these types of metabolic enzymes as biomarker candidates for studying dystrophinopathy-related changes in bioenergetic pathways. The comparable levels of fibronectin in wild type versus dystrophin-deficient EOMs indicates the lack of reactive myofibrosis in this type of muscle, which is otherwise seen in most contractile tissues affected in the dystrophic phenotype [28,33,67]’.    
Revised lines 428-440: ‘A potential weakness and bioanalytical limitations of this report relate to the usage of an animal model of Duchenne muscular dystrophy instead of patient samples, and the focus on changed protein concentration using peptide mass spectrometry. With the exception of the diaphragm, the mdx-4cv mouse exhibits relatively mild symptoms of fibre wasting in its general musculature. Thus, findings from this study should ideally be extended to characterize muscle specimens from Duchenne patients, which is however extremely difficult in the case of EOMs. Proteomics focuses on the mass spectrometric identification of individual proteoforms and can be routinely employed for comparative studies. However, it is important to realize that the establishment of abundance changes in individual proteins does not provide detailed information on the underlying regulatory mechanisms. It will therefore be important to supplement the new proteomic data sets with future analyses on the biochemical, physiological and cell biological properties of dystrophin-deficient EOMs’. 

Reviewer 2, Comment 5: ‘Figure 5. The quality of the top plates from EOM mdx-4cv in Eosin and MyHC14 needs to be significantly improved, even that of plate wt in Eosin.
Response:  We agree and have substituted the histological images with new H&E images and also replaced the immunofluorescence labelling of MyHC14 in mdx-4cv EOM sections with a new image. Due to the Covid crisis our laboratories had to work under extremely difficult conditions including limited access to equipment, restricted shift work of lab personnel and the discontinuation of the mouse colony used for this study. This resulted in limitations in the amounts of tissue being available for verification analyses. However, we believe that the displayed immunoblotting analyses and immunofluorescence microscopical studies are sufficient to verify key findings of the proteomic survey of EOMs. 

Revised figure legend of new Figure 5: ‘Figure 5. Histological and immunofluorescence microscopical characterization, as well as immunoblot analysis, of extraocular muscle (EOM) from the mdx-4cv mouse model of Duchenne muscular dystrophy. Shown are transverse cryosections of wild type (wt) and mdx-4cv EOMs stained with haematoxylin and eosin (H&E) and labelled with antibodies to glyceraldehyde-3-phosphate dehydrogenase (GAPDH) and myosin heavy chain MyHC14. Bar equals 30μm. Shown is an InstantBlue Coomassie stained protein gel with wild type EOM (lane 1) and mdx-4cv EOM (lane 2) samples, as well as identical nitrocellulose replicas used for immunoblotting and labelled with antibodies to glyceraldehyde-3-phosphate dehydrogenase (GAPDH) and fibronectin (FN). In the below panels are shown the statistical analysis of immunoblotting (Mann-Whitney U test; n=4; *p<0.05). The value of molecular mass standards (x10-3 kDa) is marked on the left side of the gel.

Revised text in relation to new Figure 5 (lines 397-411): ‘Immunofluorescence microscopy and immunoblotting was employed to compare the expression of glyceraldehyde-3-phosphate dehydrogenase. As illustrated in Figure 5, immunofluorescence labelling of this glycolytic enzyme shows elevated levels in dystrophic EOMs, as compared to relatively comparable levels of myosin MyHC14. Immunoblotting also indicates increased levels of glyceraldehyde-3-phosphate dehydrogenase in mdx-4cv EOMs, as compared to comparable expression of fibronectin. An approximately 2-fold increase in the expression of glyceraldehyde-3-phosphate dehydrogenase has previously been identified in dystrophic mdx-23 diaphragm preparations [86]. Thus, a potential shift to more glycolytic metabolism appears to be associated with the dystrophic phenotype, and this seems to be especially striking in mdx-4cv EOMs, suggesting these types of metabolic enzymes as biomarker candidates for studying dystrophinopathy-related changes in bioenergetic pathways. The comparable levels of fibronectin in wild type versus dystrophin-deficient EOMs indicates the lack of reactive myofibrosis in this type of muscle, which is otherwise seen in most contractile tissues affected in the dystrophic phenotype [28,33,67]’.  

Reviewer 3 Report

Gargan et al. use mass spectrometric analysis to compare proteome in extraocular muscles (EOMs) and diaphragm and to compare the EOM proteome in wild-type and mdx4cv mice. The rationale behind the study is that EOMs are functionally unaffected in Duchenne muscular patients (DMD) in the course of the disease progression. In contrast, the phenotype in patients suffering from DMD is very severe in the diaphragm. Therefore, the comparable proteomic analyses of the muscles are justified. The authors then follow the mass spectrometric analyses with immunoblotting, confirming high levels of myosin MHC14 in the wt EOM muscles (compared to wt diaphragm) and immunofluorescence microscopy showing high levels of GAPDH in mdx4cv EOM (compared to wt EOM). Although descriptive, this is a very well-presented study that should be of interest to the broader audience.

I have only a few minor remarks:

1) Introduction:

I think the authors should mention that there are various dystrophin isoforms. While Dp260 is specifically present in EOMs, diaphragm contains Dp427. These isoforms differ in terms of their capability to interact with the cytoskeleton.

2) Introduction:

The authors should mention that Mdx4cv mice lack not only Dp427 but also Dp260.

3) p. 3; Experiments for verification analysis, the establishment of multi-consensus files and comparative proteomics were performed with n=8, n=6 and n=3 specimens from wild type and dystrophic mice, respectively.

Please rewrite so that it is clear how many mice of each genotype were used for a particular analysis.

4) p. 3

There is no information regarding antibodies and concentrations that were used for immunoblotting and immunofluorescene microscopy.

5) Immunofluorescence microscopy showing high levels of GAPDH in the mdx4cv EOM could be followed by immunoblotting. This could be particularly interesting in the light of one of your older papers (Doran et al. 2006), where only a 2-fold increase in the GAPDH level was noted in the mdx diaphragm. In contrast, the increase noted for mdx4cv EOM is 18-fold.

6) Discussion:

The authors should discuss in more detail their results, in term of various MyHC isoforms and their possible functions as well as the function of dystrophin in EOMs vs diaphragm.  

Author Response

Reviewer 3, Comment 1: ‘Gargan et al. use mass spectrometric analysis to compare proteome in extraocular muscles (EOMs) and diaphragm and to compare the EOM proteome in wild-type and mdx4cv mice. The rationale behind the study is that EOMs are functionally unaffected in Duchenne muscular patients (DMD) in the course of the disease progression. In contrast, the phenotype in patients suffering from DMD is very severe in the diaphragm. Therefore, the comparable proteomic analyses of the muscles are justified. The authors then follow the mass spectrometric analyses with immunoblotting, confirming high levels of myosin MHC14 in the wt EOM muscles (compared to wt diaphragm) and immunofluorescence microscopy showing high levels of GAPDH in mdx4cv EOM (compared to wt EOM). Although descriptive, this is a very well-presented study that should be of interest to the broader audience. I have only a few minor remarks: 1) Introduction: I think the authors should mention that there are various dystrophin isoforms. While Dp260 is specifically present in EOMs, diaphragm contains Dp427. These isoforms differ in terms of their capability to interact with the cytoskeleton’.
Response: We agree and now mention the different isoforms of dystrophin in the revised Introduction section, as follows.
Revised lines 64-69: ‘... The extremely large DMD gene contains 7 different promoters that are involved in the tissue-specific expression of several small dystrophin isoforms, i.e. Dp45 (central nervous system), Dp71-G (ubiquitous), Dp116-S (Schwann cells), Dp140-B/K (brain and kidney) and Dp260-R (retina), as well as the large dystrophins, i.e. Dp427-M (skeletal muscle, heart muscle, smooth muscle), Dp427-B (brain) and Dp427-P (Purkinje cells) [23]’.

Reviewer 3, Comment 2: ‘2) Introduction: The authors should mention that Mdx4cv mice lack not only Dp427 but also Dp260’.
Response: We would like to thank reviewer 3 for pointing out this important characteristic of the mdx-4cv mouse model of dystrophinopathy. The dystrophin status of the mdx-4cv model is now mentioned in the revised manuscript as follows:
Revised lines 114-116: ‘… The harvesting of post-mortem EOMs and diaphragm muscle specimens from 12-month old wild type mice and the mdx-4cv mouse model of dystrophinopathy, which is lacking the dystrophin isoforms Dp140, Dp260 and Dp427 due to a point mutation in exon 53, was carried out according to institutional regulations’.

Reviewer 3, Comment 3: ‘3) p. 3; Experiments for verification analysis, the establishment of multi-consensus files and comparative proteomics were performed with n=8, n=6 and n=3 specimens from wild type and dystrophic mice, respectively. Please rewrite so that it is clear how many mice of each genotype were used for a particular analysis’.
Response: The revised Methods section now states the requested information as follows:
Revised lines 119-122: ‘… The establishment of multi-consensus files was carried out with muscle samples from 6 wild type and 6 dystrophic mice. Comparative proteomics was performed with specimens from 3 wild type versus 3 dystrophic mice. Verification analyses were carried out with samples derived from a minimum of 4 wild type and 4 dystrophic mice.

Reviewer 3, Comment 4: ‘4) p. 3, There is no information regarding antibodies and concentrations that were used for immunoblotting and immunofluorescene microscopy’.
Response: This information has been added to the revised manuscript as follows:
Revised lines 179-183: ‘… For immunoblotting, gel electrophoretically separated proteins were transferred to nitrocellulose membranes, blocked in fat-free milk solution and incubated in 1:1000 diluted primary antibody overnight. The subsequent detection with 1:1000 diluted peroxidase-conjugated secondary antibodies was carried out with the enhanced chemiluminescence method’.
Revised lines 194-201:  ‘… at room temperature. Primary antibodies to myosin heavy chain MyHC14 and glycer-aldehyde-3-phosphate dehydrogenase were diluted 1:200 and 1:400, respectively, in carrageenan-containing and phosphate-buffered saline for overnight incubation at 4°C. The buffer was made by mixing 100 ml phosphate-buffered saline with 0.7 g carrageenan and 10 mg sodium azide. Tissue specimens were carefully washed and then incubated with fluorescently labelled secondary antibodies, using 1:500 diluted anti-mouse RRX antibody for 45 min at room temperature. Nuclei were counter-stained with …’.

Reviewer 3, Comment 5: ‘5) Immunofluorescence microscopy showing high levels of GAPDH in the mdx4cv EOM could be followed by immunoblotting. This could be particularly interesting in the light of one of your older papers (Doran et al. 2006), where only a 2-fold increase in the GAPDH level was noted in the mdx diaphragm. In contrast, the increase noted for mdx4cv EOM is 18-fold’.
Response: In response to this point, we have carried out additional immunoblots with the small amounts of remaining EOM tissue preparations available for this study. Revised Figure 5 now contains the immunoblotting of GAPDH and fibronectin in wild type versus mdx-4cv EOMs. In contrast to relatively comparable levels of fibronectin, the immunoblotting of the GAPDH protein indicates increased levels of this glycolytic enzyme in dystrophin-deficient EOMs.
Revised figure legend of new Figure 5: ‘Figure 5. Histological and immunofluorescence microscopical characterization, as well as immunoblot analysis, of extraocular muscle (EOM) from the mdx-4cv mouse model of Duchenne muscular dystrophy. Shown are transverse cryosections of wild type (wt) and mdx-4cv EOMs stained with haematoxylin and eosin (H&E) and labelled with antibodies to glyceraldehyde-3-phosphate dehydrogenase (GAPDH) and myosin heavy chain MyHC14. Bar equals 30μm. Shown is an InstantBlue Coomassie stained protein gel with wild type EOM (lane 1) and mdx-4cv EOM (lane 2) samples, as well as identical nitrocellulose replicas used for immunoblotting and labelled with antibodies to glyceraldehyde-3-phosphate dehydrogenase (GAPDH) and fibronectin (FN). In the below panels are shown the statistical analysis of immunoblotting (Mann-Whitney U test; n=4; *p<0.05). The value of molecular mass standards (x10-3 kDa) is marked on the left side of the gel.

Revised text in relation to new Figure 5 (lines 397-411): ‘Immunofluorescence microscopy and immunoblotting was employed to compare the expression of glyceraldehyde-3-phosphate dehydrogenase. As illustrated in Figure 5, immunofluorescence labelling of this glycolytic enzyme shows elevated levels in dystrophic EOMs, as compared to relatively comparable levels of myosin MyHC14. Immunoblotting also indicates increased levels of glyceraldehyde-3-phosphate dehydrogenase in mdx-4cv EOMs, as compared to comparable expression of fibronectin. An approximately 2-fold increase in the expression of glyceraldehyde-3-phosphate dehydrogenase has also been identified in dystrophic mdx-23 diaphragm preparations [86]. Thus, a potential shift to more glycolytic metabolism appears to be associated with the dystrophic phenotype, and this seems to be especially striking in mdx-4cv EOMs, suggesting these types of metabolic enzymes as biomarker candidates for studying dystrophinopathy-related changes in bioenergetic pathways. The comparable levels of fibronectin in wild type versus dystrophin-deficient EOMs indicates the lack of reactive myofibrosis in this type of muscle, which is otherwise seen in most contractile tissues affected in the dystrophic phenotype [28,33,67]’.  

Reviewer 3, Comment 6: ‘6) Discussion: The authors should discuss in more detail their results, in term of various MyHC isoforms and their possible functions as well as the function of dystrophin in EOMs vs diaphragm’.
Response: We agree and have added the requested discussion points to the revised manuscript, as follows:
Revised lines 308-321: ‘The EOM-specific expression and distribution of specialized types of myosin heavy chains, such as MyHC-13, MyHC-14 and MyHC-15 [17], which surround the globe structure within the ocular cavity, could be related to the unusual physiological properties of this type of muscle. In addition, the broad isoform expression pattern in EOMs including both slow and fast myosin isoforms, i.e. slow MyHC-I, fast MyHC-IIa, fast MyHC-IIb and fast MyHC-IId, as well as embryonic MyHC-3 and perinatal MyHC-8, form probably the functional basis for an extremely wide range of possible eye movements. This allows complex movements of the eyeball in relation to its horizontal and vertical axis ranging from slow vergence to rapid saccades [6]. EOMs are capable of both considerable eccentric contraction patterns and fibre twitching at high frequency without tetanus, and this is at least partially provided by the kinetic properties of the unique combination of myosins and their distribution  in EOMs. For example, the embryonic MyHC-3 isoform is located mostly to the terminal region of contractile fibres, while the superfast MyHC-13 isoform is positioned in a region at the central endplate [62]’.
Revised lines 355-374: ‘The relatively mild phenotype of EOMs in X-linked muscular dystrophy [19,20] is probably closely related to the special biochemical and physiological features of the muscles surrounding the eyeball, such as the longitudinal distribution of neuromuscular junctions [9-11], the considerable capacity for fibre regeneration [13,14], exceptional fa-tigue resistance even in fast-twitching fibre populations [16] and an extensive calcium extrusion system [15]. In most skeletal muscles, dystrophin deficiency causes a collapse of sarcolemmal integrity and a concomitant increase in micro-rupturing of the surface membrane, which in triggers influx of Ca2+-ions into myofibres and associated enhanced Ca2+-dependent proteolytic activity in the sarcosol [25,27]. The efficient and swift removal of cytosolic calcium from EOM fibres and the enhanced ion buffering capacity of EOMs might therefore play a key role in the protection from dystrophic changes [18,75,78]. Another important factor might be the relatively low concentration of dystrophin isoform Dp427-M in EOMs. The sub-sarcolemmal dystrophin lattice might not play the same crucial role in the membrane cytoskeleton and linkage of the intracellular actin filaments to extracellular laminin via the dystrophin-glycoprotein complex as in other skeletal muscles [36]. Importatntly, previous studies have established an up-regulation of the autosomal dystrophin homologue utrophin Up-395 and associated rescue of sarcolemmal glycoproteins such as beta-dystroglycan in dystrophin-deficient EOMs [37,76,77]. Thus, full-length utrophin might substitute for dystrophin in EOMs and thereby stabilize its trans-sarcolemmal cytolinker function and prevent secondary damage to myofibres’.

Round 2

Reviewer 2 Report

Thank you for the revision, it significantly improved the quality of the manuscrip.

However, my major concern regarding the possibility of having included the retractor bulbi muscle has not been addressed in detail. Please state whether care has been taken not to include the retractor bulbi muscle, located behind the eye, surrounding the optic nerve and inside the EOM cone.

Author Response

Reviewer 2 comment: ‘Thank you for the revision, it significantly improved the quality of the manuscript. However, my major concern regarding the possibility of having included the retractor bulbi muscle has not been addressed in detail. Please state whether care has been taken not to include the retractor bulbi muscle, located behind the eye, surrounding the optic nerve and inside the EOM cone.

Response: Thanks for your positive reply. Our preparation has excluded the retractor bulbi muscle, and this information has now been added to the re-revised text in the Methods section (lines 122-125): ‘The eyeball and its surrounding tissues were carefully removed from the ocular cavity by bulbar exenteration. The EOM cone excluding the retractor bulbi muscle was then dissected out and extracted for the isolation of the combined EOM proteome’.